# Optical Imaging Opportunities to Inspect the Nature of Cytosolic Iron Pools

**DOI:** 10.3390/molecules28186467

**Published:** 2023-09-06

**Authors:** Robert Charles Hider, Charareh Pourzand, Yongmin Ma, Agostino Cilibrizzi

**Affiliations:** 1Institute of Pharmaceutical Science, King’s College London, London SE1 9NH, UK; 2Department of Life Sciences, University of Bath, Bath BA2 7AY, UK; prscap@bath.ac.uk; 3Centre for Therapeutic Innovation, University of Bath, Bath BA2 7AY, UK; 4Centre for Bioengineering and Biomedical Technologies, University of Bath, Bath BA2 7AY, UK; 5Institute of Advanced Studies, School of Pharmaceutical and Chemical Engineering, Taizhou University, 1139 Shifu Avenue, Taizhou 318000, China; yongmin.ma@zcmu.edu.cn

**Keywords:** labile iron pools, fluorescence, iron sensitive optical probes, near-infrared probes, imaging

## Abstract

The chemical nature of intracellular labile iron pools (LIPs) is described. By virtue of the kinetic lability of these pools, it is suggested that the isolation of such species by chromatography methods will not be possible, but rather mass spectrometric techniques should be adopted. Iron-sensitive fluorescent probes, which have been developed for the detection and quantification of LIP, are described, including those specifically designed to monitor cytosolic, mitochondrial, and lysosomal LIPs. The potential of near-infrared (NIR) probes for *in vivo* monitoring of LIP is discussed.

## 1. Introduction

Iron-dependent proteins play a central role in oxygen transport, electron transport, redox reactions, and nucleotide synthesis. Iron is typically incorporated into proteins as heme, iron sulfur clusters, or directly into tertiary structure cavities [1]. As there is a heavy demand for iron in order to maintain normal metabolic processes, iron is efficiently conserved in the mammalian body. Indeed, mammalian iron excretion is highly controlled, and the iron content of sweat and urine is low. It has been estimated that once absorbed, the average iron atom spends approximately 10 years inside a human before being excreted. Thus, iron is located in a range of intracellular pools, some relatively kinetically inert such as ferritin and others, such as the “labile iron pool” (LIP), which is labile, thereby facilitating the transmembrane movement of iron and iron donation to apoenzymes. Bioavailable, cytosolic iron exists in the iron(II) (ferrous) state and loosely interacts with a range of counter ions [2]. Williams argued that the electrode potential of the cytosol favours iron(II) over iron(III), and because the iron(II)-binding constants for many cytosolic enzymes fall in the range 10^−8^–10^−7^ M [3], it requires a similar standing concentration of iron(II) in order to prevent dissociation from the enzyme. Fluorescent probe studies, using calcein [4] and Phen Green SK [5], have demonstrated that the majority of the cytosolic LIP is iron(II), which, in view of the 100-fold higher kinetic lability of iron(II) when compared with iron(III) [6], is a logical strategy. Iron(II) is generally the form involved in intracellular translocation, for instance, incorporation into iron-requiring enzymes, incorporation into ferritin, and transport across membranes by the divalent metal transporter 1 (DMT1) and ferroportin. This kinetically labile group of iron species, LIP, is estimated to exist in concentrations of the order of 1 µM [2].

Iron(II) is capable of forming a range of oxygen radicals under aerobic conditions, via Fenton chemistry [7,8] and so the level of LIP needs to be carefully controlled, thereby limiting the potential toxicity of this critically important iron pool. In mammals, cellular LIP homeostasis is achieved by two “Iron Regulatory Proteins” (IRP 1 and IRP 2), which control the iron uptake into the cell via transferrin receptor 1, as well as iron export via ferroportin and iron storage in ferritin. This network of feedback mechanisms is presented schematically in Figure 1. Both IRP 1 and IRP 2 are RNA-binding proteins, capable of binding to ‘iron response elements’ (IREs) on mRNA [9]. By binding to IREs they inhibit the translation or degradation of mRNAs, which encode proteins involved in iron storage and iron import, thereby influencing LIP levels [10]. With IRP 1, IREs compete for binding to the protein with the coenzyme 4Fe-4S cluster of aconitase (ACO 1). With IRP 2, the insertion of a di-iron centre into the F-box and leucine-rich repeat protein 5 (FBXL5) permits the formation of a ubiquitin ligase complex, which subsequently binds to IRP 2, inducing its breakdown by the proteasomal pathway. Thus, the assembly of these two iron centres, namely 4Fe-4S and Fe-O-Fe, creates an integrated iron sensing mechanism for LIP, which in turn is linked to many feedback loops.

There are additional iron-sensing networks that are adopted in a wide range of organisms; another well-characterized example is based on iron- and oxygen-dependent prolyl hydroxylases [11]. The addition of a hydroxyl group to specific prolines within the Hypoxia-Inducible Factor (HIF), a transcription factor involved in hypoxia and iron metabolism, leads to their ubiquitination and subsequent breakdown [11]. The activity of prolyl hydroxylases is iron-dependent and so LIP levels influence the cytosolic levels of HIF.

## 2. Chemical Components of LIP

Many ligands previously associated with the cytosolic LIP chelate iron(III) but not iron(II) at pH 7.0; namely, amino acids [12], ATP/AMP [13], and inositol phosphates [14]. In contrast, citrate does bind iron(II) under cytoplasmic conditions and has been suggested to be the major component of LIP [15]. However, Fe^2+^·citrate is susceptible towards autoxidation at pH 7.0 [16], which renders citrate an unlikely iron(II) buffer. Based on the relatively strong interaction between iron(II), thiol-containing compounds, cysteine, and glutathione (GSH) have also been considered as possible cytosolic ligands for iron(II) [17]. Speciation plots based on the typical cytosolic levels of cysteine indicate that this ligand does not bind iron(II) sufficiently tightly to make a significant contribution to the LIP [17]. In contrast, GSH is present in the cytoplasm at a much higher concentration than that of cysteine, with the cytosolic level of GSH in human erythrocytes being 2.5 mM [18] and in rat liver being 8 mM [19]. Using the logK_1_ value for iron(II) of 5.12 [17], GSH is predicted to bind iron(II) to a considerable extent at pH 7.0 (Figure 2). GSH also rapidly reacts with *hexaaquo*·iron(III) converting it to *hexaaquo*·iron(II) (**1**) (Figure 1) [20]. 

GSH does not chelate iron(II) as, unlike cysteine, the bidentate coordination would require the formation of chelate rings containing either 9, 10, or 11 atoms. Such ring sizes are associated with unfavourable entropy differences. The non-chelating structure (**2**) (Figure 1) agrees with that previously proposed for glutathione coordination of metal ions [21].

Over the concentration ranges of GSH (2–10 mM) and iron(II) (0.5–5 µM), the two major cytosolic iron(II) species are reliably predicted to be Fe^2+^·GS (**2**) and *hexaaquo*·Fe^2+^ (**1**), with the glutathione complex dominating. The concentration of [Fe(H_2_O)_6_]^2+^ (**1**) is linearly related to the total concentration of cytosolic iron(II) and is predicted to vary between 10^−8^ M and 5 × 10^−7^ M for a range of total iron(II) concentration between 1 µM and 10 µM. Clearly, the large excess of glutathione does not prevent [Fe^2+^]_cytosol_ from acting as a potential signal for iron(II) sensors, such as the IRP 1/aconitase- and IRP 2/FBXL5-linked systems. Because of the high kinetic lability of iron(II), this system will be close to equilibrium and therefore at pH 7 the speciation plot depicted in Figure 2 will accurately reflect the cytosolic concentrations. Although autoxidation of cytosolic Fe^2+^·GS will occur at a slow rate, the resulting iron(III) will be rapidly reduced back to iron(II) (Figure 3), in a fashion similar to that of the reduction in methemoglobin. 

Iron(II) binding by GSH offers a means by which the cytosol can distinguish iron(II) and manganese(II), both of which are present at similar levels [22]. The affinity of manganese(II) for GSH is markedly lower than that of iron(II), with the logK_1_ values being 2.7 [23] and 5.1 [17], respectively. This difference has a major effect on Mn(II) speciation, where citrate is found to be the dominant ligand, with GSH failing to bind even a trace of Mn(II) under cytoplasmic conditions [17]. 

As iron(II)glutathione, under physiological conditions, exists as a hydrated iron(II) cation with a single coordination site being occupied by monodentate glutathione (**2**), other low-molecular-weight ligands present in the cytosol can simultaneously coordinate to this ion, forming a ternary complex. The additional bidentate or tridentate ligand would need to possess a similar affinity for iron(II) as that of GSH in order to form an appreciable concentration of the ternary complex; the ligand would also need to be present at a relatively high concentration (1–10 mM). These requirements severely limit the number of such possibilities. Significantly, a group of widely distributed histidine-containing dipeptides, including carnosine (**4**) (Figure 1), are capable of binding transition metals [24]. Indeed, it has been suggested that carnosine can compete for iron(II) with hypoxia-inducible factor 1 alpha (H1F-1α), an iron-dependent proline hydroxylase [25]. Furthermore, carnosine has previously been demonstrated to form mixed transition metal complexes with glutathione [26]. A likely structure for such a ternary complex of carnosine, glutathione, and iron(II) is indicated by **3** (Figure 1).

Thus, LIP is likely to consist of several iron(II) complexes in equilibrium with each other (Figure 2), with glutathione acting as a redox buffer locking iron into the iron(II) state. The concentrations of iron(II), glutathione, and carnosine will influence the relative concentrations of the LIP components (**1**), (**2**), and (**3**). In such a situation, it is important to establish which form of LIP generates an intracellular signal. Is this task limited to *hexaaquo*·iron(II) (**1**) or could GS·Fe^2+^ (**2**) and GS·Fe^2+^·carnosine (**3**) trigger some more specific intracellular events? 

In all probability, it is only *hexaaquo*·Fe^2+^ (**1**) that provides the critical intracellular signal.

## 3. The Nature of the Labile Iron Pool in Different Intracellular Organelles

***Mitochondrial Labile Iron Pool***. Mitochondria are major sites for heme and iron-sulfur cluster synthesis in both plants and animals and, consequently, there is a constant iron influx into mitochondria as they export both heme and iron-sulfur clusters for use elsewhere in the cell. There are almost certainly multiple mechanisms for mitochondrial iron uptake. Mitoferrins 1 and 2 have, for instance, been linked to the delivery of iron to mitochondria in a range of organisms [27,28], with the accumulation being dependent on the mitochondrial membrane potential [29]. Significantly there is also a considerable movement of glutathione across mitochondrial membranes, facilitated by the dianionic exchangers; malate^2−^/HPO_4_^2−^ and 2-oxoglutarate^2−^/malate^2−^ [30,31]. An interesting possible variant of such an exchange is for Fe^2+^·GS (**3**) to act as a substrate for this family of transporters. Indeed it has been suggested that iron(II) is transported into the mitochondrion as a complex rather than as inorganic iron [29].

The concentrations of citrate and glutathione are higher in the mitochondrial matrix than in the cytosol; namely [citrate] = 1.1–1.25 mM [32,33] and [GSH] = 11 mM [19]. Speciation analysis confirms that under these conditions, Fe^2+^·GS is the dominant form of iron(II) at pH 8.0 [2]. Indeed, a pH of 8.0 further favours the formation of Fe^2+^·GS (**2**) when compared to the cytosol. Furthermore, the mitochondrial ratio of [GSH]/[GSSG] is higher than that found in the cytosol [34]. 

In addition to an iron(II) buffering role, Fe^2+^·GS binds to glutaredoxins, proteins that are required for iron cluster assembly and heme biosynthesis [35]. Glutaredoxins are small proteins that possess a glutathione-binding site and a redox-active thiol function, both of which are located on the surface of the protein. These thiol functions typically catalyse thiol-disulphide exchange reactions. Some glutaredoxins are also capable of simultaneously binding iron and glutathione, forming [2Fe-2S] clusters, which are shared between two subunits of a homodimer [36,37]. These [2Fe-2S] clusters are transferred to acceptor proteins and are also involved in iron-regulatory protein (IRP) regulation [38]. The mitochondrion is the dominant location for iron-sulfur cluster synthesis [39], but the precise steps in [4Fe-4S] cluster biosynthesis are not completely defined [40], although glutathione has been demonstrated to possess a central role in the process [41,42]. The overall biogenesis occurs in two parts: The de novo assembly of a Fe-S cluster on a scaffold protein, such as the Iron-Sulfur Cluster Assembly Enzyme (ISCU), and then the subsequent transfer to target apoproteins. Glutaredoxins and glutathione are directly involved in these processes [36,43,44]. In the presence of glutathione, [2Fe-2S] clusters undergo a reversible exchange between apo-ISCU and glutathione, forming a complex [45]. Reductive coupling of two [2Fe-2S] clusters to form a single [4Fe-4S] cluster takes place on homodimeric cluster scaffold proteins [44] and the resulting cluster can, in turn, be transferred to apoproteins. Thus, it is conceivable that glutathione acts not only as a ligand for mononuclear iron(II) but also for [2Fe-2S] clusters.

***Lysosomal Labile Iron Pool.*** The lysosome is involved in the breakdown of many iron-containing proteins, in particular ferritin. The digestion of ferritin generates a pool of ferric iron, which, when exposed to the acid pH range of most lysosomes (pH 4.5–5.5), will be complexed by citrate. Citrate levels in the lysosome typically fall in the range of 10–20 mM. Glutathione has a relatively low affinity for iron at pH 5.0 (Figure 2) and so the concentration of iron(II)glutathione is predicted to be much lower than that found in the cytosol. Thus, the predominant iron species in the lysosome is likely iron(III) coordinated with citrate.

Iron(III) citrate and other related complexes are capable of redox cycling in the lysosome and this can be reduced by intraliposomal iron chelation [46]. Iron(III) citrate is also a substrate for CytB reductase, reducing iron(III) to iron(II), thereby generating a substrate for DMT1. Thus, lysosomal LIP is recycled into the cytosol LIP in a controlled manner. 

***Nuclear Labile Iron Pool.*** The sequestering of genetic material within the nucleus of the eukaryotic cell provides the nucleus with the ability to regulate gene expression. Iron is an essential component for the expression of many genes [47,48] and of multiple enzymes involved in DNA metabolism [49], including both DNA synthesis and repair. Iron-dependent enzymes include DNA polymerase and primase, DNA-helicases, nucleases, glycosylases, and demethylases together with ribonucleotide reductases, many of which contain iron sulfur clusters [49]. 

The boundary between the nucleus and cytoplasm is a double membrane termed the nuclear envelope. This envelope is perforated with nuclear pore complex structures through which nucleocytoplasmic transfer occurs [50,51]. All passive transport across the nuclear envelope is through the nuclear pore complexes, which possess a central aqueous channel 9 nm in diameter. This channel permits the passive movement of molecules measuring less than 50 KDa [50]. Macromolecules larger than 50 KDa require combined assistance from nuclear localization signals and the nuclear pore complex. Thus, unless there are some extremely efficient transmembrane ion and solute pumps present in the nuclear envelope, the cytosolic and nucleus solute concentrations will be similar. Thus, *hexaaquo*·iron(II) and iron(II)glutathione are likely to be the major labile species in the nuclear pool. 

## 4. Isolation and Characterization of Labile Iron Species

It should be clear from the previous discussion in this overview that in all probability, the LIP consists of a number of components and that these may differ between intracellular compartments by virtue of different pH values and concentrations of various iron-binding compounds in these various compartments. In the majority of compartments, namely cytosol, the mitochondrion, and the nucleus, the dominant redox state is iron(II), which is kinetically labile. This property renders it extremely difficult to isolate and purify such species from any biological matrix. Chromatography will not be possible due to the lability of iron(II)glutathione; iron dissociates from the complex and binds to other competing ligands including the stationary phase of the column. Mass spectrometry would appear to be the most promising method for the identification of labile iron species in biological matrices [52]. We are currently investigating such possibilities. At the present time, the most successful procedures for the quantification of LIP are fluorescent-based assays.

## 5. Iron-Sensitive Fluorescent Probes

The design of iron probes is typically based on bifunctional structures containing both a chelating agent and a fluorophore. A third function can be incorporated into the probe in order to influence subcellular localization [53,54]. For iron, the chelating agent can be selective for either iron(II) or iron(III). Probes to detect iron may contain either “turn-on” fluorophores, in which the binding of iron increases fluorescence, or “turn-off” fluorophores, in which the binding of iron quenches the fluorescence [55]. Although “turn-on” probes are preferred because they lead to more accurate quantification of iron, they have proven to be difficult to design and, at present, most currently available fluorescent iron probes are of the “turn-off” type. The chelation of iron by a probe leads to dynamic fluorescence quenching as both the quantum yield and the lifetime of the fluorescence decrease [56].

With ‘turn off’ probes, the concentration of LIP is determined by the decrease in fluorescence after correcting for photobleaching and leakage of the probe using either an *in situ* or an *ex-situ* calibration method [5,57]. Based on Equation (1), the equilibrium constant (K_probe_) can be used to quantify LIP, where LIP = [Fe^2+^] + [Fe^2+^·probe]. In principle, K_probe_, [Fe^2+^·probe], and [probe] can be determined. In order to minimise the efflux from the cell, the acetoxy methoxy group (AM) has been successfully introduced. AM enhances the membrane permeability of the probe, but cytosolic esterases rapidly cleave the AM esters to the more hydrophilic carboxylate function (Equation (2)). As a result, the now hydrophilic probe remains trapped within the cytosol [58].
(1)
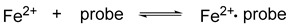

(2)
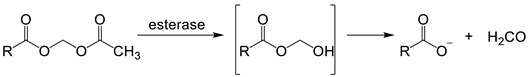

(3)
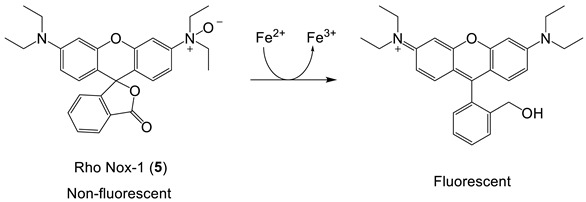


Rho Nox-1, an interesting iron(II) probe, has been investigated for the fluorescent-based detection of LIP [59,60]. The Fe^2+^-mediated deoxygenation of the N-oxide group on the fluorophore is reported to lead to an enhanced fluorescence, and as such, it is a “turn-on” probe, Equation (3). The mechanism of this deoxygenation reaction is not clear at the present time, and the reduction in the N-oxide may also be facilitated by thiols, including glutathione, which is present in the cytosol at high concentrations. More control studies are required before this class of probes can be reliably used to report LIP concentrations. 

***Cytosolic Probes.*** There are many fluorescent probes, selective for either iron(II) or iron(III), that have been previously described [61,62]. However, the properties of many of these probes have not been carefully investigated in cell culture systems. We discuss probes that have been investigated in some detail, namely calcein-AM (**6**), Phengreen SK diacetate (**7**), Fura-2AM (**8**), Rho Nox-1 (**5**), and Cytosense LI^TM^.

*Calcein-AM* (**6**) is currently the most widely adopted fluorescent iron probe, having been studied in a wide range of cells [63,64]. Typical behaviour is presented in Figure 4 where a single HepG2 cell is monitored [65]. As iron(II), presented as iron(II) ammonium sulfate, enters the cytosol it quenches the calcein fluorescence. Upon the addition of Salicylaldehyde isonicotinoyl hydrazone (SIH, **9**) (40–100 µM), the chelator gains rapid access to the cytosol where it out-competes calcein for iron(II) and consequently the fluorescence rapidly increases. When the behaviour of SIH (**9**) and deferoxamine (DFO) are compared in the presence of another cell line, it is clear that DFO fails to gain entry to the cytosol during the time scale of the study Figure 5 [63].



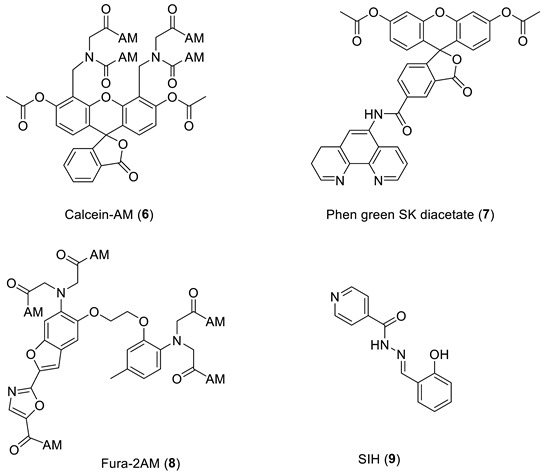



As indicated by the discussion centred on Equation (1) using such probes, it is, in principle, possible to determine the concentration of LIP. The concentration of **6** can be determined from the fluorescence of a single cell or a suspension of cells. [Fe^2+^·**6**] is measured from the change in fluorescence upon the addition of SIH and the binding constant of iron(II) to the probe. However, the relevant binding constant for calcein is not certain; iron(II) certainly binds to the carboxylate functions of calcein, but physical measurements have demonstrated that iron also binds to phenolic groups on calcein, which can favour autoxidation of iron(II) to iron(III) [66]. Thus there is uncertainty regarding the chemical structure of the physiologically relevant iron complex that acts as a sink; indeed, a mixture of complexes likely form and thus uncertainty exists as to what iron(II)-binding constant to use. An additional limitation of calcein is that it is a substrate for a xenobiotic exporter protein, which facilitates leakage from cells.

Despite this limitation, calcein has been utilized extremely effectively to provide information on changes in LIP levels in a range of cell types. Thus, the rate of influx of iron chelators into cells can be directly compared [67] (Figure 6), the UVA-induced production of LIP can be monitored [68] (Figure 7), and the influence of LIP on the erythrocyte stage of *Plasmodium falciparum* can be monitored during cell culture (Figure 8) [69].

*Phen green SK diacetate* (**7**) is metabolized by cytosol esterases to Phen green SK (**10**), as shown in Equation (4), and hence is largely trapped in the cytosol. However, a small proportion of **10** gains access to the mitochondria and lysosomes [70]. Phen green SK diacetate (**7**) has been utilized to measure cytosolic LIP by using an *ex-situ* calibration method [70,71]. When this method was applied to isolated rat hepatocytes, a range of LIP values was obtained (Figure 9), with a large proportion falling in the range of 0.8–2.0 µM [71].
(4)
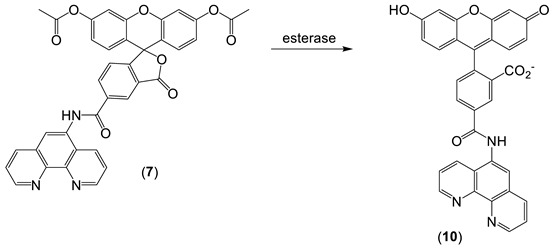


*Rho Nox-1* (**5**) is reported to be selectively sensitive to the presence of iron(II) [59], and confocal fluorescence microscopy images of HepG2 cells incubated with (NH_4_)_2_Fe(SO_4_)_2_ register an increase in fluorescence intensity (Figure 10). As indicated in the earlier section on “Iron-sensitive fluorescent probes”, more control studies on this group of probes are urgently required. Stability studies are reported for coincubation with GSH, but only at 1 mM, although cytosolic GSH is at least 8 mM. It has not been established whether Rho Nox-1 (**5**) can detect iron(II) in the presence of physiological levels of GSH.

*Hydroxypyridin-4-one (HOPO)*-containing probes have been investigated for the quantification of LIP [54]. Under biological conditions, they are selective for iron, binding iron(II), and rapidly autoxidizing the metal to iron(III) (Equation (5)) in a similar fashion to that of DFO. CP655 (**11**) achieves rapid entry into cells without the necessity of using AM or acetyl esters. However, this ability to penetrate plasma membranes provides CP655 access to both lysosomes and mitochondria, thus providing a measure of cellular LIP as opposed to cytosolic LIP. A further limitation of this class of fluorescent probes is that they are not trapped intracellularly and will efflux once the incubation media is changed [72]. However, changes in LIP levels induced by the addition of iron(III) citrate and a competing chelator CP94, in a similar fashion to those reported for calcein and Phen green SK, can be monitored by changes in fluorescence (Figure 11).
(5)
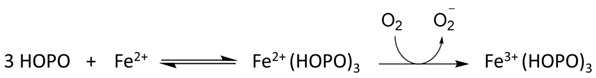




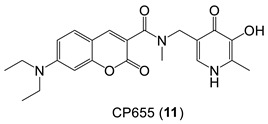



**Figure 11 molecules-28-06467-f011:**
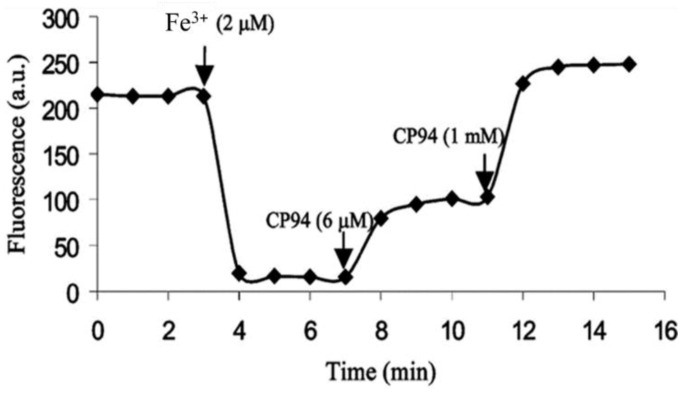
Fluorescence of the hydroxypyridinone CP655 (6 μM) in primary hepatocytes, quenched by iron(III) (2 μM) and recovered in the presence of CP94 [72].

In principle, HOPO probes, which are linked to ester moieties, could be targeted to the cytosol in a similar manner to that of calcein AM (**6**). We have synthesized a range of such molecules and, as a result, identified *Cytosense LI* as being suitable for the quantification of cytosolic LIP. Time-lapse fluorescent microscopy demonstrates that *Cytosense LI* enters cells rapidly and responds to the addition of iron presented as a complex with 8-hydroxyquinoline (Figure 12). The determination of the concentration of LIP in primary skin fibroblast FEK4 cells was achieved by measuring the fluorescence difference of pairs of samples, namely *Cytosense LI* and DFO and conversion to [LIP] using an ex situ calibration curve (Figure 13). A value of 0.29 ± 0.02 µM was obtained for a mean of four determinations [73]. More control studies are required for *Cytosense LI*, but it appears to be a promising probe for cytosolic LIP measurement.

*Endoperoxide reactivity-based probes* have also been utilized to provide an estimate of cytosolic iron(II) levels [74,75]. They are reported to be selective for iron(II) via the Fenton reaction. However, to date, there have been no further reports on the application of this class of probes for monitoring iron(II) levels.

***Mitochondrial Probes.*** As mitochondria are the main utilizers of iron in the cell, as well as being a major source of the superoxide anion, O_2_^−•^, iron homeostasis is critically important. An increase in the mitochondrial iron burden is likely to be linked to associated pathology, for instance with Friedreich’s ataxia where there is inefficient iron sulfur cluster synthesis [76]. The mitochondrial LIP is buffered to some extent by mitochondrial ferritin, but not in all tissues, for instance, liver mitochondria lack mitochondrial ferritin.

One widely adopted method of targeting molecules to the mitochondrial matrix is by attachment to delocalized lipophilic cations. By taking advantage of the substantial negative electrochemical potential maintained across the inner mitochondrion membrane (typically −200 mV), delocalized cations are able to cross the membrane and hence be accumulated with a distribution driven by the Nernst potential, typically in excess of 1000-fold. Triphenyl phosphonium salts (**12**) are such examples where the positive charge formally associated with the central phosphorus atom is delocalized over the three phenyl rings, resulting in a low electron density. This, together with the overall lipophilic character of the molecule, enables rapid penetration of membranes. This useful property of lipophilic cations has been utilized to target fluorescent iron sensors in the mitochondrion. Both Petrat’s [77] and Cabantchik’s groups [78,79] have utilized a derivative of rhodamine (RPA, **13**) to monitor mitochondrial iron levels. Rhodamine is a positively charged fluorescent molecule where the single positive charge is delocalized on both nitrogen atoms and over the plane of the tricyclic structure. By adopting methods developed for the cytosol, estimates of the mitochondrial LIP were made. The values for mitochondria of different cells were found to be distributed over a wide concentration range, centred around 12 µM [77]. A limitation of this probe is the difficulty in reversing the quenched signal under physiological conditions [79] and thus the associated difficulty of unequivocally assigning the quenched signal to labile iron.



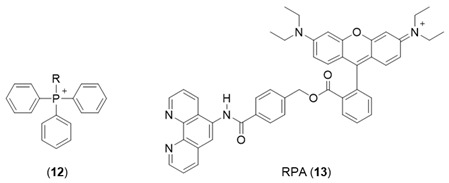



Largely as a result of the above limitation, a different approach has been introduced for monitoring the mitochondrial matrix LIP. A series of peptides have been prepared, members of which are capable of being targeted to and accumulated in mitochondria. These peptides contain alternate hydrophobic and basic amino acids and typically contain between 4 to 6 residues. Using this general design, a series of small fluorescent peptides have been synthesized that are selectively accumulated by mitochondria [80]. By incorporating an iron-chelating moiety into the molecule (**14**), it becomes possible to monitor the mitochondrial LIP. BP19 (**14**) has been utilized to monitor the mitochondrial LIP of cultured fibroblasts obtained from Friedreich’s ataxia patients [81]. The mean levels of mitochondrial LIP in Friedreich’s ataxia cells were found to be 1.11 ± 0.37 µM, whereas those from healthy donors were 0.17 ± 0.12 µM. This finding supports the observation of Rouault et al. [82] that there is an accumulation of redox-active labile iron in the mitochondria of Friedreich’s ataxia patients.



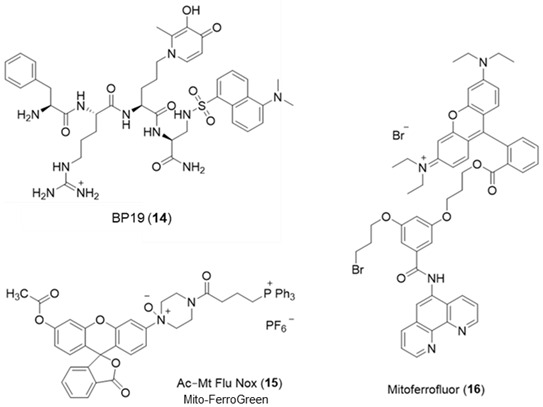



A mitochondria-targeted, NO-containing fluorescent probe Mito-FerroGreen (**15**) has been introduced for the measurement of mitochondrial LIP [83]. It contains the triphenyl phosphonium entity to facilitate mitochondrial targeting and an NO function, which renders the fluorescence of the probe iron(II) sensitive. The same limitations exist with this molecule as with *Rho Nox-1* (**5**), namely more control studies are required to establish iron(II) selectivity. Another phenanthroline-based mitochondrial probe Mitoferrofluor (**16**) has been investigated for the detection of iron(II) [84]. This probe is also quenched by copper (II), but not at normal physiological copper levels. This probe can detect mitochondrial chelatable Fe^2+^ in both polarized and depolarized mitochondria.

***Lysosomal targeting Probes.*** The endosomal/lysosomal compartment of cells contains a relatively high level of labile iron, largely resulting from the breakdown products of both mitochondria [85] and ferritin [86]. Chelation of this iron pool protects against oxidative stress-induced cellular damage [46,87]. The measurement of labile iron levels in the endosomal/lysosomal compartment of hepatocytes is difficult because of its small size, approximately 1% of the cell volume [88]. However, the larger lysosomes present in liver endothelial cells are amenable to measurement. Using Phen Green SK in combination with 1,10-phenanthroline, Petrat and coworkers determined a labile iron level for this intracellular compartment of 15.8 ± 4.1 µM [88].

The hydroxypyridinone SF34 (**17**) has been demonstrated to only be located in endosomal/lysosomal compartments in macrophages [89]. SF34 does not permeate membranes rapidly and so can be trapped within the endosomal/lysosomal compartment, which it enters by endocytosis. This probe has been utilized to compare the ability of different chelators to mobilize lysosomal LIP using single-cell flow cytometry [90].



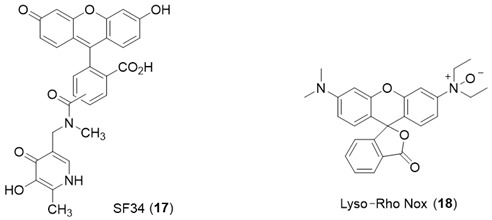



The *Rho Nox-1* analogue, *Lyso-Rho Nox* (**18**), has been reported to be localized in the lysosomal compartment of human adenocarcinoma (MCF-7) cells, and this property has been utilized to report differences in lysosomal LIP levels [91].

## 6. NIR Probes

The development and application of near-infrared (NIR) technologies is also under investigation, in order to facilitate our understanding of the nature of cytosolic iron pools *in vivo*. This is primarily due to the unique properties of NIR light and its interaction with biological tissues. Namely, by exploiting a region of the electromagnetic spectrum, which is less absorbed and scattered (i.e., above 650 nm), NIR light is known to offer several advantages, such as increased tissue penetration, limited autofluorescence by endogenous molecules, and, thus, minimized background interference, along with reduced photobleaching of the fluorophores and photodamage to the tissues. All this leads to an enhanced signal-to-noise ratio by NIR agents and, thus, clearer and more reliable imaging results for *in vivo* imaging. In this context, NIR-based agents represent promising tools for the study of intracellular iron levels in organs and tissues, which would be challenging or impossible to access with shorter-wavelength probes (e.g., emitting in the ultraviolet-visible region). Iron-targeted NIR probes have been developed with this purpose in mind [92,93,94] to facilitate *in vivo* investigations of iron-related disorders, including neurodegeneration and other syndromes by the imbalance of iron homeostasis. Typically, these probes are designed in such a way that ensures high specificity for iron, exhibiting a change in their fluorescence properties upon binding to the metal, thereby enabling the monitoring of its dynamics in the cytosol with enhanced sensitivity and spatial resolution. Noteworthily, the ability to selectively detect iron(II) and/or iron(III) species is of great importance to the understanding of the roles of this metal in biological processes. This has the potential to facilitate the identification of new and tailored therapeutic interventions for diseases characterized by iron-overload or, in general, altered physiological levels of iron. To date, a limited number of NIR probes for iron(III) have been developed, and there is only a single report on the application of iron(II)-sensitive NIR probes for *in vivo* imaging [94,95,96]. 

We present below a selection of Fe-targeted NIR probes that would hold promise for further development in the field and possible clinical translation, such as DCI-Fe(II) (**19**), CAM (**20**), LS1 (**21**), and DFO_m_-NPs (**22**) (Figure 14). However, other agents with iron sensing mechanisms not fully defined or difficult to be adopted in *in vivo* settings have also been reported [97,98,99,100,101], along with bioluminescent and supramolecular systems [102,103], as well as technologies based on 2D materials, quantum dots, and metal-organic frameworks [104,105,106]. 

In the context of iron(II)-targeted imaging, Zheng and collaborators developed the fluorescent probe DCI-Fe(II) (**19**) (Figure 14), which can sense the metal on the basis of *N*-oxide chemistry [107], being reported to be highly effective in the detection and imaging of iron(II) ions in real-time, both in cells and *in vivo*. In particular, this agent demonstrated a metal concentration-dependent NIR signal turn-on response (at 700 nm) coupled to a large Stokes shift (195 nm), along with a detection limit of 51 nM for iron(II). However, the selectivity of this probe for iron(II) remains to be established as indicated earlier in this review. Additionally, a far-red probe (i.e., 620 nm emission) was reported as an effective imaging probe for the detection of Fe(II) levels in mouse models of inflammation [101]. In a similar fashion, a caged D-aminoluciferin construct was recently developed as a bioluminescent reporter for visualizing iron pools in luciferase-expressing cells and mouse models, working through an iron-dependent uncaging mechanism, which is reported to enable sensitive and selective imaging of ferrous over ferric species [96,102].

With regard to iron(III) probes for *in vivo* imaging, Zhu et al. reported the synthesis of a cyanine-based NIR agent (**20**, CAM) (Figure 14) equipped with two *N*-(2-hydroxyethyl)acetamide groups to specifically chelate iron [108]. This agent functions as a “turn-off” probe, providing a decrease in fluorescence emission at 804 nm upon binding to iron(III) ions. Moreover, the probe demonstrated very high selectivity for iron(III) ions over a range of other metal ions, with a detection limit of approximately 8 μM, although its suitability for *in vivo* imaging would require further assessment in animal models. Li et al. introduced the iron-targeted NIR probe LS1 (**21**) (Figure 14), which utilises a cyanine-based fluorophore attached to the rhodamine B dye [109]. This probe is reported to produce a ratiometric response in detecting iron(III) ions, although it can also sense Cu(II) levels. Upon the addition of iron(III), the fluorescence intensity of **21** (Figure 14) at 777 nm decreases, while there is an increase in the intensity at 577 nm, which is due to the iron(III)-induced spirocyclic ring opening followed by the activation of a cyanine-mediated photoinduced electron transfer process. As with the NO-based probes, selectivity for iron needs to be carefully established before this agent (**21**) can be extensively used for *in vivo* imaging. In 2019, Kang and colleagues reported NIR probes coupled with deferoxamine (DFO) [110]. The design of these agents focuses on engineered bio-conjugation strategies to incorporate DFO into a backbone of ε-poly-L-lysine residues (DFO_m_-NPs, **22**) (Figure 14), which in turn is attached to the zwitterionic NIR fluorophore ZW800-1C, in order to enable effective *in vivo* tracking. Analogues of such probes could have the potential for the detection and quantification of LIP *in vivo*. 

These examples demonstrate the diversity and potential of NIR probes for the measurements of iron levels *in vivo*. The non-invasive nature and high sensitivity of these probes may offer valuable insights into the complexities of iron homeostasis and metabolism, as well as its implications in health and disease. As the field of bioimaging continues to advance, more elaborate and/or effective probes are likely to be developed.

## 7. Conclusions

This review provides information on the chemical nature of LIP and outlines the function of this important metabolic pool. The possible variation of LIP chemistry in the cytosol, mitochondria, lysosomes, and nucleus is discussed. The precise chemical structures of the major components need to be identified in biological matrices and this is likely to be achieved by ICP-MS techniques.

The potential of using iron-sensitive probes for the quantification of LIP is considered, and a range of applications of such probes is described. The quantification of LIP by the use of iron-specific fluorescent probes is currently under development, but it is now possible to monitor LIP levels in cytosolic, mitochondrial, and lysosomal pools of cells in suspension. Such measurements made under *in vivo* conditions will be more difficult, but NIR probes are under development for this purpose. Undoubtedly, the ability to visualize and quantify iron dynamics in living organisms is expected to have a significant impact on the development of targeted therapeutic interventions for iron-related pathological conditions.

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
