# Peer review of "Optical Imaging Opportunities to Inspect the Nature of Cytosolic Iron Pools"

_molecules, 2023, doi:10.3390/molecules28186467_

Round 1

Reviewer 1 Report

The authors have written an excellent review article explaining the detection issues of intracellular labile iron pools. The article is clear and perfectly organized. The authors precisely describe the chemical transformations of iron(II) and iron(III) ions in the cellular environment, which makes LIP detection quite complicated. The article presents the latest reports on detecting iron ions in different intracellular organelles with the aid of various fluorescent probes.

I strongly recommend this article to Molecules.

Author Response

We thank Referee 1 for their feedback and positive comments on our manuscript.

Reviewer 2 Report

This is an interesting and comprehensive review of the field of optical imaging of cellular labile iron pools with a strong focus on the underpinning chemistry of this area. It opens with a useful discussion of the chemistry and types of labile ion pools found in cells that is thorough and clearly explained. It then covers the various fluorescent probes currently in use and under development targeting organelles such as mitochondria drawing on an extensive range of up to date literature sources. The section on NIR probes is particularly interesting as it is an emerging area. Overall this review will interest both the users and developers of these iron probes and I a recommend its publication in its present form.

Author Response

We thank Referee 2 for their feedback and positive comments on our manuscript.

Reviewer 3 Report

The chemical nature of intracellular labile iron pools (LIPs) is described. And detection of LIP in cytosolic, mitochondrial and lysosomal pools of cells by iron-specific fluorescent and NIR probes were overviewed, which providing us  an imaging view to inspect the nature of cytosolic iron pools.

There are several issues need to be revised or discussed:

1. The symbols of iron(II), Fe2+, FeII and Fe(II) should be consistent;

2.  The sections concerning NIR probes were too simple and should be supplemented.

Author Response

We thank Referee 3 for their feedback and comments on our manuscript. Our reply to their queries follows.

  1. The symbols of iron(II), Fe2+, FeII and Fe(II) should be consistent;

We have revised this aspect and improved the uniformity between word/symbols within the manuscript, including in scheme 2, figures 2, 11, and equations 1 and 5. The only forms that we now use in the text are iron(II)/iron(III) or Fe2+/3+ in the case that these are part of a chemical formula.  

  1. The sections concerning NIR probes were too simple and should be supplemented.

As suggested, we have expanded this section with additional details and literature records.

Reviewer 4 Report

Robert Charles Hider et al. in the submitted manuscript “Optical imaging opportunities to inspect the nature of cytosolic iron pools” summarize scientific literature describing the chemical nature of intracellular LIPs, iron-sensitive fluorescent probes for the detection of LIPs and the potential of NIR probes for monitoring LIPs. Overall, the manuscript is a useful summary of research papers that published about iron-specific fluorescent probes. Therefore, I am happy to recommend this manuscript to be accepted for publishing in molecules after the major revision:

1.        One of the review’s aims is highlighting developments of fluorescence probes for monitoring of LIP, but do not provide a summary table which would include basic properties (absorbance maximum, extinction coefficient, emission maximum, quantum yield, fluorescence lifetime, charge and etc.) of these probes. In light of this, it is highly recommended that such a table be incorporated into the manuscript. The inclusion of such a table would considerably enhance the manuscript's utility, as potential readers are likely to find these parameters crucial when selecting a suitable probe for their experimental needs.

2.        Further enhancement of the aesthetic quality of the images in the article is required. Additionally, certain figures appear to lack clarity, such as Figure 2, Figure 7 and Figure 8.

3.        The notation "hexaaquo·iron(II) (1)" in line 90 should be designated as belonging to Scheme 1 to enhance readability and mitigate potential reader misinterpretation. Moreover, chemical structures and equations presented within the article should be accompanied by appropriate figure captions.

4.        The conclusion section is notably succinct. It is recommended that the conclusion begin by providing a concise summary of the principal content of the article, followed by an exposition of the deficiencies extant within the field and prospects for future development.

5.        The manuscript contains several formatting errors. I kindly request a comprehensive review of the entire manuscript and effect necessary revisions. For instance, within the abstract, the author states: "Iron-sensitive fluorescent probes which have been developed for the detection and quantification of LIP are described, including those specifically designed to monitor, cytosolic, mitochondrial, and lysosomal LIPs." There should not be a comma between "monitor" and "cytosolic". Furthermore, a period is missing between the title of the referenced work [69] and the name of the journal in the citation section. Moreover, the eighth reference in the citation section lacks bold formatting for the year; and so forth.

6.        Before being considered for publication in MDPI molecules, I would encourage the authors to address some critical points listed below, which are related to referencing previous contributions of research reports in this area:

An Endoperoxide Reactivity-Based FRET Probe for Ratiometric Fluorescence Imaging of Labile Iron Pools in Living Cells (DOI: 10.1021/jacs.6b08016); A reactivity-based probe of the intracellular labile ferrous iron pool (DOI: 10.1038/NCHEMBIO.2116); In vivo bioluminescence imaging of labile iron accumulation in a murine model of Acinetobacter baumannii infection (DOI: 10.1073/pnas.1708747114); Learning from Artemisinin: Bioinspired Design of a Reaction-Based Fluorescent Probe for the Selective Sensing of Labile Heme in Complex Biosystems (DOI: 10.1021/jacs.9b11245); Molecular Imaging of Labile Heme in Living Cells Using a Small Molecule Fluorescent Probe (DOI: 10.1021/jacs.1c08485).

none comments

Author Response

We thank Referee 4 for their feedback and comments on our manuscript. Our reply to their queries follows.

  1. One of the review’s aims is highlighting developments of fluorescence probes for monitoring of LIP, but do not provide a summary table which would include basic properties (absorbance maximum, extinction coefficient, emission maximum, quantum yield, fluorescence lifetime, charge and etc.) of these probes. In light of this, it is highly recommended that such a table be incorporated into the manuscript. The inclusion of such a table would considerably enhance the manuscript's utility, as potential readers are likely to find these parameters crucial when selecting a suitable probe for their experimental needs.

Although this is a good suggestion, at the present time it is not possible to produce such a table. Most of the probes are based on standard fluorophores (e.g. carboxyfluoresceine, dansyl, coumarin). However, we are not certain that the optical properties have been published for the final probes.

  1. Further enhancement of the aesthetic quality of the images in the article is required. Additionally, certain figures appear to lack clarity, such as Figure 2, Figure 7 and Figure 8.

We have revised all figures and adopted visual improvements for Figure 2, Figure 7 and Figure 8.

  1. The notation "hexaaquo·iron(II) (1)" in line 90 should be designated as belonging to Scheme 1 to enhance readability and mitigate potential reader misinterpretation. Moreover, chemical structures and equations presented within the article should be accompanied by appropriate figure captions.

We have linked hexaaquo-iron(II) to Scheme 1 in line 90. We would prefer to leave the chemical structures 6-9 and 11-18 unassociated with any figure. This is typical practice for chemical publications. We do not think it is appropriate to give figure captions to equations.

  1. The conclusion section is notably succinct. It is recommended that the conclusion begin by providing a concise summary of the principal content of the article, followed by an exposition of the deficiencies extant within the field and prospects for future development.

As recommended, the conclusion section has been modified and implemented following the reviewer’s suggestions.

  1. The manuscript contains several formatting errors. I kindly request a comprehensive review of the entire manuscript and effect necessary revisions. For instance, within the abstract, the author states: "Iron-sensitive fluorescent probes which have been developed for the detection and quantification of LIP are described, including those specifically designed to monitor, cytosolic, mitochondrial, and lysosomal LIPs." There should not be a comma between "monitor" and "cytosolic". Furthermore, a period is missing between the title of the referenced work [69] and the name of the journal in the citation section. Moreover, the eighth reference in the citation section lacks bold formatting for the year; and so forth.

We have revised the manuscript and corrected formatting mistakes and typos throughout, including the suggestions in the abstract and in ref 8. We consider reference 69 to be correctly presented.

  1. Before being considered for publication in MDPI molecules, I would encourage the authors to address some critical points listed below, which are related to referencing previous contributions of research reports in this area:

An Endoperoxide Reactivity-Based FRET Probe for Ratiometric Fluorescence Imaging of Labile Iron Pools in Living Cells (DOI: 10.1021/jacs.6b08016); A reactivity-based probe of the intracellular labile ferrous iron pool (DOI: 10.1038/NCHEMBIO.2116); In vivo bioluminescence imaging of labile iron accumulation in a murine model of Acinetobacter baumannii infection (DOI: 10.1073/pnas.1708747114); Learning from Artemisinin: Bioinspired Design of a Reaction-Based Fluorescent Probe for the Selective Sensing of Labile Heme in Complex Biosystems (DOI: 10.1021/jacs.9b11245); Molecular Imaging of Labile Heme in Living Cells Using a Small Molecule Fluorescent Probe (DOI: 10.1021/jacs.1c08485).

We have considered these papers and included in the manuscript the 1st and 2nd paper from those suggested by the reviewer. In addition, we have modified the text to refer to these papers at the end of the section on cytosolic probes. The article “In vivo bioluminescence imaging of labile iron accumulation in a murine model of Acinetobacter baumannii infection” was already cited in the original version of the manuscript; please old ref. [100] (i.e. [102] in the revised version). The last 2 suggested articles are on detection of labile heme. Thus, although the obvious links between heme and iron, we consider these beyond the scope of this review.

Round 2

Reviewer 4 Report

This review is ready for publication